# An All-Solid-State Coaxial Structural Battery Using Sodium-Based Electrolyte

**DOI:** 10.3390/molecules26175226

**Published:** 2021-08-28

**Authors:** Federico Danzi, Pedro Ponces Camanho, Maria Helena Braga

**Affiliations:** 1LAETA, Engineering Physics Department, Engineering Faculty, University of Porto, 4200-465 Porto, Portugal; 2INEGI, Institute of Science and Innovation in Mechanical and Industrial Engineering, 4200-465 Porto, Portugal; pcamanho@fe.up.pt; 3LAETA, Mechanical Engineering Department, Engineering Faculty, University of Porto, 4200-465 Porto, Portugal

**Keywords:** structural batteries, ferroelectric, sodium solid electrolyte, composite materials, multifunctional materials

## Abstract

The transition to a sustainable society is paramount and requires the electrification of vehicles, the grid, industry, data banks, wearables, and IoT. Here, we show an all-solid-state structural battery where a Na^+^-based ferroelectric glass electrolyte is combined with metallic electrodes/current collectors (no traditional cathode present at fabrication) and thin-ply carbon-fiber laminates to obtain a coaxial multifunctional beam. This new concept aims to optimize the volume of any hollow beam-like structure by integrating an electrochemical system capable of both harvesting thermal and storing electrical energy while improving its mechanical performance. The coaxial cell is a coaxial cable where the dielectric is ferroelectric. The electrochemical results demonstrated the capability of performing three-minute charges to one-day discharges (70 cycles) and long-lasting discharges (>40 days at 1 mA) showing an energy density of 56.2 Wh·L^−1^ and specific energy of 38.0 Wh·kg^−1^, including the whole volume and weight of the structural cell. This is the highest specific energy among safe structural cells, while no Na^+^-based structural cells were found in the literature. The mechanical tests, instead, highlighted the coaxial cell capabilities to withstand severe inelastic deformation without compromising its functionalities, while increasing the flexural strength of the hosting structure. Moreover, the absence of alkali metals and liquid electrolytes together with its enhanced thermal properties makes this coaxial structural battery a valid and safe alternative as an energy reservoir for all the applications where traditional lithium-ion batteries are not suitable.

## 1. Introduction

Battery technologies, especially lithium-ion (LIB), have revolutionized the trend for all electrical applications in the last decades. Their combination of high energy and power density made them a ubiquitous solution for portable electronic devices, vehicles, and many other applications. Nonetheless, the batteries in use, such as the LIB, do not charge fast enough, are not safe due to their flammable electrolyte, and are constituted by cathodes in trends of reaching their theoretical capacity and, therefore, evolving from engineering steps.

The LIB cannot be used freely at temperatures above 40 °C due to the risk of thermal runaways and oxygen releases.

In this context, research centers from all over the world are trying to push forward this field and realize greener and more efficient battery systems. While most of the industries are focusing on optimizing the electrochemical performance and the safety of the LIB [1,2], in recent years, several research groups fostered the idea of extending batteries’ functionalities to incorporate more than one primary function [3]. Among the various ideas that emerged to obtain multifunctional batteries, one of the most promising was the possibility to endow batteries with mechanical capabilities designated by structural batteries [4,5,6,7,8,9,10].

The advantage of these batteries is related to the allocation of their volume and weight to elements that pre-exist with structural functions. The overall weight of the battery is then reduced, and the stability of the structure is highly enhanced. Starting from this concept, several designs for structural power composites have already been proposed. The pioneering works performed by Thomas et al. [4,11,12] for unmanned aerial vehicles (UAV) and maritime applications demonstrated the possibility of integrating conventional LIB in sandwich structures without critically deteriorating the panel mechanical response. More recently, Galos et al. [13,14,15] made an extensive study on the integration of Li-polymer batteries in composite laminates, showing their possible application in the automotive industry. An engineered version of the battery integration technology in carbon fiber laminates was proposed by Ladpli et al. [16], who enhanced the mechanical properties of the laminates with embedded cells using interlock rivets.

A completely different approach was followed by Asp and coworkers [8,9], who pioneered the production of multifunctional power composite materials. Once the possibility of using the carbon fiber as anode for a structural battery was demonstrated [17,18,19], Hagberg et al. [20] studied the possibility of coating LiFePO_4_ (LFP) particles on the surface of the fiber to use them as a cathode. The coated and uncoated fibers were then integrated as reinforcement in a biphasic polymeric resin impregnated with a liquid electrolyte to guarantee the electrochemical properties. This new concept of electrolyte, in which the liquid electrolyte is an ionic liquid, named structural battery electrolyte [21,22], was employed to manufacture the prototypes of a full multifunctional power composite cell [23] with remarkable malfunction properties while showing reduced specific energy.

Another possible hybrid approach was proposed by Moyer and co-workers [24,25], who designed a composite plate with carbon-fiber-reinforced plastics (CFRP) face sheets encapsulating a carbon fiber-based Li-ion battery. The cell, in this case, was fabricated with doped graphite as anode and a cathode both soaked in an ionic liquid electrolyte with a commercial separator to avoid short circuits.

Lately, Pyo et al. [26] presented a tubular laminated composite battery manufactured with a hybrid glass/carbon fiber shell package that contains an organic liquid electrolyte-based battery. The cell was produced with LiFePO_4_ (LFP)- and Li_4_Ti_5_O_12_ (LTO)-coated aluminum film electrodes interleaved by a glass fiber separator soaked with LiPF_6_ liquid electrolyte. While the LTO anode is safer than graphite, this battery was still using a flammable liquid electrolyte.

Table 1 summarizes the main characteristics and performance of the structural batteries described in the recent literature including their constituents, the manufacturing process, the specific energy, the operating voltage, and highlights the use of liquid electrolytes in structural batteries, from which those having a traditional flammable liquid electrolyte show higher specific energies.

Notwithstanding the remarkable results already achieved by the previously presented approaches, several issues must be overcome: (1) the use of the liquid flammable electrolyte, which is critical for application in structural elements that may be exposed to high mechanical load and structural failure; (2) the application of this multifunctional element to a limited temperature range; (3) the use of carbon fibers/graphite as anodes enduring lithiation/delithiation leading to a quick degradation and eventual delamination; (4) the necessity for a complex battery management system (including temperature sensors) limiting its applications; (5) the cost.

It is noteworthy that all the commercial cylindrical batteries, except the alkaline Zn-MnO_2_, difficultly allow for a solid-state electrolyte in replacement of the liquid; layers of the anode-separator-cathode, soaked in liquid electrolyte, constitute them. The introduction of a solid-state electrolyte separator would most likely result in electrolyte/separator cracks and delamination, especially while enduring mechanical stress. 

Here, an all-solid-state electrolyte was applied to structural batteries, enhancing the mechanical properties of the cell and avoiding using pre-existing cathodes and therefore preventing carbon fibers degradation upon cycling. Also presented is a ferroelectric Na^+^-based electrolyte coaxial cell (see Figure 1a), which shows the highest specific energy among structural cells with safe electrolytes. The coaxial shape facilitates cell fabrication and potentiates a higher capacitance and better mechanical performance with reduced delamination risk.

It is highlighted that, because of this battery design, the CFRP outer cylinder shell can be independently cured with one of the electrodes in compliance with the material supplier recommendation without mining the cell mechanical performance, even in the case of high temperature curing cycles. This aspect, as shown in the recent literature, places relevant limitations in the integration of battery components with high-performance CFRP structures since it forces the use of low-temperature curing cycle resins and adhesives [14,24] or the filling of the cured composite with liquid electrolyte that must be carried out in a glovebox with safety concerns and limitations of the structural dimensions [16,23,26].

This configuration allows for a sequential and flexible manufacturing process that can be carried out without any need for a glovebox. Moreover, the thick solid layer of electrolyte allows to completely remove the separator without any risk of a short circuit even in the presence of severe inelastic deformations.

The cells can be easily associated in series and be made in any dimension. Their open circuit voltage is given by the difference in electrochemical potentials of the electrodes and the cells can be charged in 3 min at higher currents.

Moreover, the enhanced thermal response of the coaxial structural battery design makes it extremely attractive to be used not only as a structural element but also for thermal energy and waste heat harvesting device in industries, databanks, photovoltaics, buildings, and vehicles. Figure 1 shows only some of the possible applications for the structural battery proposed here.

## 2. Materials and Methods

### 2.1. Fabrication of the Cells

The peculiarity of the composite coaxial structure relies on the use of a ferroelectric glass electrolyte with enhanced thermal and electrochemical properties enabled by its ferroelectric and ionic character [27,28].

As introduced in Figure 2, the coaxial structural batteries are composed of three elements, an outer CFRP-based cylindrical shell that hosts the copper foil acting as pre-cathode/current collector and fixing the positive electrode’s chemical potential, the all-solid-state electrolyte, and the coaxial anode/current collector fixing the chemical potential of the negative electrode.

In this study, the total length of the coaxial beam was fixed to 100 mm, while the value of the internal radius of the outer shell was set to 6 mm. The metallic anode, instead, was cut 20 mm longer than the full coaxial beam, while a small prolongation was kept on the cathode/collector foil to allow the electrical connection. The outer shell of the CFRP beams, including the copper positive electrode, was obtained by wrapping and curing the 50 × 100 mm^2^ laminates around a cylindrical aluminum beam with an outer diameter of 12 mm. The layup used in this study is a quasi-isotropic [90/0/ +45/ −45]_S_ fabricated using the T800-736LT 100 gsm thin-ply carbon-epoxy prepreg supplied by NTPT [29]. On the inner side of the composite cylinder, a 37 × 95 mm^2^ rectangular foil of copper with a thickness of 0.025 mm from Alfa Aesar was placed, aligning one of its short sides with the short side of the CFRP laminate. The carbon fiber shell and the cathode/current collector foil were then co-cured in the hot press for 9 h at 80 °C.

The dry glass Na^+^ solid electrolyte with enhanced ferroelectric and thermoelectric properties was synthesized by water solvation, as presented by Braga et al. [30]. The all-solid-state electrolyte Na_2.99_ Ba_0.005_OCl_1-x_(OH)_x_ was obtained from the commercial precursors NaCl (99.0%, JMS, HongKong, China), Na(OH) (98.0%, LabKem, Dublin, Ireland), and Ba(OH)_2_·8H_2_O (98.5%, Merck, Kenilworth, NJ, USA). The precursors were mixed with deionized water before permitting them to react and dry between 230 °C and 250 °C.

Once it reached its final solid state, the electrolyte was ground using a ball milling machine with an Agate container and balls with a diameter of 20 mm, which can be closed hermetically to avoid water absorption during the process. The electrolyte powder was eventually mixed with the thermoplastic Polyvinyl Acetate (PVAc) (C_4_H_6_O_2_)_n_ that was determined to be the sole constituent detected in the mixture electrolyte and white glue used (0.8 Na_2.99_Ba_0.005_OCl + 0.2 PVAc) (see the scanning electron microscopy SEM micrographs in Figure 3 and the X-ray photoelectron spectra XPS spectra in Supporting Materials, Appendix A). The mixture was rendered to improve the hygroscopic character while maintaining the dielectric properties of the electrolyte, and to promote the electrical and mechanical adhesion between the electrodes and the electrolyte of the composite coaxial beam. The mixture can be made homogeneous, as shown in Figure 3.

The negative electrode was fabricated by cutting an aluminum rod with a radius of 2 mm. The coaxial batteries were then fabricated placing the negative electrode in the center of the outer cell using a dedicated tool and filling the gap between the electrodes with the solid-state electrolyte composite. The electrolyte mixture in the gap was finally hand-compacted in the coaxial structure to guarantee the contact between all the elements of the coaxial beam. Finally, the edges of the structural battery were sealed with thermoplastic glue to prevent the electrolyte from absorbing moisture.

### 2.2. Electrochemical Analysis of the Cells

The all-solid-state coaxial structural batteries were then evaluated both electrochemically and mechanically to characterize their multifunctional performance.

Electrochemical impedance spectroscopy (EIS) was used to estimate the internal resistance of the cells while a series of cyclic voltammetry (CV) experiments, at different voltage rates, was performed to estimate their capacitance. The three-point bending study of the cells was performed to determine their failure load while monitoring the batteries’ electrical performance behavior under load. Figure 4 shows the various types of coaxial cells developed and analyzed in this work. For the sake of completeness, on the left, a glass fiber version of the proposed structural batteries has been presented to show the internal structure of the cells while, on the right, the basic CFRP circular beam is presented.

Impedance spectroscopies (EIS) analyses were performed to determine the internal resistance of the cells (Figure 5a). The tests were performed using a Gamry Reference 3000 potentiostat, and the cells were kept in a sandbox container at a constant temperature of 40 °C. All the experiments were carried out with an alternate current AC with an amplitude of ±10 mV for an initial differential potential corresponding to the OCV of the cell. The frequencies varied from 1 MHz down to 200 mHz. From the impedance obtained, it was possible to calculate the electrolyte’s conductivity σ via Equation (1)
(1)$$R=\frac{1}{2\pi \sigma l}ln\frac{b}{a}$$
where *R* is the coaxial cell resistance to ion hopping from *a* to *b* within the electrolyte, *l* the length of the cell, *b* the inner radius of the positive electrode, and *a* the external radius of the negative electrode.

By performing cyclic voltammetry (CV) experiments, it is possible to determine the capacitance of the cells. The tests were done in the Gamry potentiostat and the structural cells were kept at 40 °C. The measurements were performed superimposing a potential difference of ± 0.2 V to the measured OCV that was fixed as 0 V (see CV curves in Appendix A). To determine the capacitance C, different scan rates ranging from 0.1 to 50 mV s^−1^ were used. The capacitance of the cell was then calculated at OCV using the *<i>* vs. *dV/dt* slope and Equation (2):(2)$$\langle i\rangle =C\frac{dV}{dt}$$
where *<i>* is the average current at OCV, *C* is the capacitance of the cell, and *dV/dt* is the applied voltage rate. From the capacitance, the dielectric constant is then calculated using Equation (3) for two coaxial cylinders separated by a dielectric material as follows:(3)$$C=\frac{2\pi \varepsilon_{r}\varepsilon_{0}l}{ln\frac{b}{a}}$$
where *ε_r_* is the dielectric constant of the cell and *ε_0_* is the permittivity of the vacuum. As time plays an important role in the polarization of the electrolyte while being cycled, the permittivity versus scan rate was determined and shown in Figure 5b. Supplementary permittivity results for an additional structural coaxial battery cell are shown in the Appendix A. 

To determine the specific energy and the cyclability of the proposed coaxial structural battery design, the cells were discharged with a constant material resistor to assure no interference from both the amplifier and the variation of the internal resistance of the cell. These analyses were carried out with a BioLogic VMP-300 potentiostat while an external resistor of 0.98 kΩ was kept connected to the cell through the discharge time of >40 days (see Figure 6). The output voltage and current of the cell were continuously monitored and the temperature was kept constant at 40 °C. The OCV of the cell is *V = (µ^−^- µ^+^)/e*, where *µ^−^* (Al) and *µ^+^* (Cu) are the chemical potentials of the negative and positive electrodes, respectively (referred to vacuum at 0 eV), and *e* is the charge of the electron. 

Results for additional long-term discharges of different structural coaxial battery cells with other material resistors 26.6 and 1.8 kΩ are shown in Figure 7a and Appendix A, respectively. 

To demonstrate the thermal properties of the proposed battery design, a cell was fully discharged at room temperature and connected to an external resistor of 1.8 kΩ. Subsequently, the output voltage and the cell temperature were monitored while raising the temperature from room temperature (22 °C) to 60 °C. The ramp-up was executed by increasing the temperature of the sandbox hosting the cell by 10 °C every hour and the values of the output recorded (see Figure 7b).

### 2.3. Mechanical Test and Cell Post-Mortem Electrochemical Analysis

Coaxial structural battery beams were mechanically tested in three-point bending and force vs. displacement curves compared with those obtained for a hollow CFRP beam with the same layup and inner diameter of the structural battery (see Figure 2b). 

The tests were performed using an Instron 4208 uniaxial testing machine equipped with a load cell of 5 kN and using a constant crosshead speed of 1.0 mm·min^−1^. Both tests were interrupted when reaching a crosshead total displacement of 16 mm. The coaxial beams, with a total length of 100 mm were placed on the support using an overhang length of 5 mm from both sides while the testing region has been equally spaced in two areas. Circular pins with a diameter of 20 mm were used both for the support and for the loading points. For the structural battery only, three 20 µm Teflon stripes were interposed in the contact points of the battery with the steel pins. This precaution was taken both to minimize the mechanical friction and avoid spurious electrical interaction between the battery and the metallic frame of the testing machine. Moreover, for the whole test, a secondary structural battery, manufactured with the same coaxial design, was connected in series to the tested one to light a green Light-Emitting Diode (LED). The use of two structural coaxial batteries is needed to overcome the 2 V potential difference required to brighten a green LED (1.83 V is the minimum voltage to light it dimly). This set-up was chosen to demonstrate the cell’s capabilities to bear severe inelastic deformation, maintaining its electrical potential difference without inducing any short circuit (see the video in Appendix A).

After the mechanical loading, the structural cell in the video in Appendix A and Figure 8 was fully unloaded and connected to the potentiostat to monitor its electrochemical properties. During a first step where the OCV of the battery was recorded, a 3-min charge at 1.3 V followed by 24 h discharge loop was imposed. After 124 h, the cycles were performed with a 1.8 kΩ material external resistor at different decreasing temperatures starting at 53 °C. 

## 3. Results and Discussion

### 3.1. Electrochemical Test Results

Results of both the EIS and the CV analyses performed on the coaxial structural batteries are presented in this work in Figure 5a. From the analysis of the results, it is possible to determine the resistance to the conduction of the Na^+^-ions in the bulk electrolyte composite of 44.0 Ω at room temperature (∼22 °C) and 5.82 Ω at 40 °C, while the resistance of the full cell is 623.6 Ω at room temperature and 138.1 Ω at 40 °C. The ionic conductivity in the electrolyte composite is then calculated using Equation (1) to be 0.41 mS·cm^−1^ at room temperature and 3.17 mS·cm^−1^ at 40 °C.

The resistance of bulk electrolyte is obtained from the first semicircle in the Cole–Cole plot in Figure 5a, as the ions in the bulk are freer to resonate at the highest applied frequencies which decrease from 1 MHz. The subsequent semicircles, at lower frequencies, reflect the resonant movement of the ions at the interfaces, where Electrical Double-Layer Capacitors (EDLCs) constituted by surface charges restrain their movement with Coulombic forces. Each semicircle in the Cole–Cole plot is represented by a capacitor in parallel with a resistor as the movement of ions in a certain direction in the electrolyte produces a negatively charged vacancy or dipole moving in the opposite direction and, therefore, the resistance of the equivalent capacitor is associated in series with it.

The resistance to the movement of the Na^+^-ions in the electrolyte decreases 7.6× by increasing the temperature from 22 to 40 °C. This variation is not just due to the insulating character of the electrolyte; it is rather observed to be related to the polarization of the electrolyte with the applied electrical field which benefits from the rising temperature; a slightly higher temperature is enough to allow the dipoles to relax and polarize more efficiently, possibly allowing the ions to move better in channels left by the alignment of the dipoles. The full cell’s internal resistance, on the other hand, decreases just 4.5× for the same temperature interval, which is expected as the resistance at the interfaces and electrodes is not expected to vary much due to this temperature difference in this range of temperatures.

The trend of the battery’s real relative permittivity was additionally obtained using CV cycles for different scan rates, as shown in Figure 5b. As the scan rate decreases, the permittivity increases, which likely reflects the ferroelectric character of the cell in which, as cycling becomes slower, the polarization takes place at a greater extent favored by a higher relaxation time. The permittivity can reach 7.5 × 10^9^ at 0.25 mV·s^−1^ and 40 °C. It is expected that optimized dimensions of the coaxial geometry favor the capacitance per comparison with the traditional double-layer flat cells, but the electrical potential scanning rate is highly reflected on the real permittivity in coaxial cells, which is in line with previous results for flat cells [30,31].

The typical discharge curve for the coaxial structural batteries shown in Figure 6 demonstrates a relevant self-charging behavior at a relatively high discharging current of approximately 1 mA, previously observed for ferroelectric pouch cells [31,32]. More in detail, Figure 6a shows the full discharge process of the battery connected to a 0.98 kΩ resistor that endured for more than 40 days. The cell started to discharge after a 48 h relaxation at OCV when a resistor was connected to the cell recording a progressive drop from 1.17 V down to 0.67 V in two days. After this discharging phase, the cell started oscillating between 0.68 V and 0.73 V for approximately 200 h before jumping up to a new equilibrium point at approximately 0.95 V. From this point, a progressive ascending trend brought the cell to 1.07 V in 29 days. The values of energy density and specific energy of 38.0 Wh·kg^−1^ and 56.2 Wh·L^−1^ were recorded after ∼42 days for the full structural cell, including the CFRP structural element (see Table 2 for further details).

Notwithstanding that the transition points for the voltage jump cannot be predicted in advance at this stage, this behavior has been systematically observed in several cells, confirming their self-charging capabilities [31,32]. It is noteworthy that the only way the electrical potential difference in a battery cell will increase while the cell is discharging is if the chemical potential of the negative electrode increases or the chemical potential of the positive electrode decreases, or both, which opposes a spontaneous discharge unless internal feedback is established as in a Poincaré two maps system, as discussed in [32], which configures an example of emergence in this complex system containing a ferroelectric electrolyte [32]. Conversely, it is not likely that the internal resistance of the battery cell decreases through a discontinuous and sharp manifestation corresponding to the voltage increase observed at approximately 300 h, but even if that occurrence was possible, it would not decrease as much as to allow for a higher voltage than the voltage of the charged cell not connected to a load. The only manner the latter decrease in internal resistance is be possible is if a negative resistance arises from the cells’ phenomena, which again configures the first scenario of feedback leading to emergence. 

Focusing more on the cell voltage trend, as shown in Figure 6b on top, these cells present an oscillating nature of their voltage and current. In the cell presented in Figure 6, the period is exactly 1 hour, as observed after Fourier transforms the voltage output signal from 450 to 470 h (see Figure 6b bottom). This phenomenon constitutes another example of a manifestation of emergence [32].

The coaxial cell in Appendix A, self-charged to a maximum of 1.16 V while submitted to a 1.8 kΩ material resistor, and the cell in Figure 7a, self-charged to 1.47 V after 3353 h (∼140 days) of being connected to a 26.6 kΩ material resistor. These latter cells show the same complex behavior as the cell in Figure 6 [31,32] and three-step increases of the voltage (1.03–1.28 V, in the first 18 days, 1.31–1.40 V in 11 days from day 82, 1.43–1.47 V in 12 days from day 112 V). The smaller discharge current (varying from 39 to 55 µA) and the self-oscillation termination around day 56, with a consequent decrease in dissipated energy (AC power), allows for a continuous self-charge process during >140 days.

The effect of the temperature on the output voltage of a cell after having been fully discharged at room temperature and kept connected to a resistor of 1.8 kΩ is shown in Figure 7b. The plot highlights how, without any applied electrical work, the cell can collect the heat from the environment to recover a voltage of ∼0.92 V that remains constant while the cell is dissipating energy on a load. The cell remained charged at room temperature after the termination of the experiment. The results reinforce the existence of a pronounced dependence of the output voltage from the temperature, especially in the range between 20 and 40 °C, in which is observable a potential difference of +0.7 V starting from 50 mV. The theoretical operating voltage of the cell is 1.1 to 1.3 V.

This intrinsic temperature-dependent mechanism is demonstrated in all ferroelectric Na^+^-electrolyte (Na_2.99_Ba_0.005_OCl)-based cells and highlights how these alternative structural batteries could be used for harvesting thermal energy and convert it into electrical energy in structures exposed to heat sources such as solar radiation, the residual heat of the industrial processes, or any other hot pipeline of HVAC or heat removal systems.

Noticeably, in a photovoltaic cell, the nominal voltage is 0.6 V and the maximum power obtained for the short circuit current is <19 mW.cm^−2^. The efficiency decreases with the increasing temperature, which can reach more than 80 °C. The latter observation may imply that this all-solid-state coaxial cell might be a viable complement or a substitute for a photovoltaic cell. Alternatively, proton-exchange membrane fuel cells (PEM) show plateau voltages that do not exceed 0.7 V, which makes the present structural type of cells viable alternatives or counterparts to PEM fuel cells in energy storage.

### 3.2. Mechanical Test Results and Post-Mortem Analysis

The force vs. displacement curves for both the coaxial structural battery and the hollow carbon fiber circular beam are shown in Figure 8a. The latter plot shows an increase of the maximum peak load for the coaxial cell of about 70% for an increase in weight of about 200% (see Table 2). The hollow beam reached a peak load of around 1.1 kN and a progressive stiffness degradation after the peak was reached. The presence of the solid-state electrolyte together with the aluminum rod coaxial insert resulted not only in a higher peak load (about 1.87 kN), but also in a pseudo-ductile region with a slight reload. The structural battery after the first drop at 1.75 kN reached a recharging phase, which is attributed to the compaction of the electrolyte in the compressed region of the beam. The load, after the second peak, remained approximatively constant until the end of the test (see the video in Appendix A). In contrast, after the peak, the hollow beam started a progressive degradation of its mechanical properties.

Both the increase in stiffness and the higher failure load were expected phenomena. Filling the hollow pipe with the compacted electrolyte and introducing an aluminum core produces a remarkable redistribution of the internal stress of the beam under bending loads that induces a change in its stiffness and its failure modes [33]. The stiffness increase is mainly related to the change in the momentum of inertia, moving from the one of a thin-walled circular beam to the one of a multilayer circular solid section. Concerning the failure modes, instead, the solid section prevents the crushing and the ovalization of the beam close to the load application points delaying the final failure. Notwithstanding that these changes in the structure behavior were expected, their quantification is not trivial since it depends on both the elastic properties of the electrolyte and the fracture properties of all the interfaces involved (CFRP/Metal/Electrolyte/Metal), whose quantification is out of the scope of this work.

It was possible to monitor the battery functionalities during the whole test thanks to an LED. The cell maintained its normal behavior without stopping lighting the LED for the whole loading phase (see the video in Appendix A). As the 16 mm displacement was reached, the load was progressively removed, and no short circuit was recorded. After the end of the test, the damaged coaxial structural battery shown in Figure 8b, with visible inelastic deformation, was still capable of providing the required current to make the LED light.

Furthermore, after the mechanical test, the coaxial structural cell was connected to the Biologic potentiostat and its open-circuit voltage was monitored for 24 h before starting a 3 min constant voltage charge at 1.3 V and 24 h discharge cycles with a 1.8 kΩ resistor. Figure 9 shows the results of this electrochemical cycling, in which it is clear that, notwithstanding the severe inelastic deformation, the cell maintained its OCV. Moreover, after 124 h, following connecting the resistor accompanied by a sharp drop of the voltage, the cell reached a minimum potential difference of 0.88 V followed by a self-charge to 1.05 V for a maximum of 1.07 V for day 62. This latter experiment shows that these novel structural cells can be pulse-charged with currents that can reach 6.5 mA, for then discharging, during a productive day (~0.6 mA), with a flat plateau voltage providing an output current defined by the external load. The experiment also shows how the cell responds to temperature drops, to then recover to the previous voltage if enough time is given for the cell to relax into a new polarized state emulating the behavior of the permittivity in Figure 5b.

The cycling protocol used in the experiment correspondent to Figure 9 configures an optimal procedure for practical applications; with very fast charges of 3 min with current reaching 6 mA at 53 °C and 1.3 mA at 42 to 40 °C, for 24 h discharges at approximately ~0.6 mA avoiding self-cycling while allowing self-charge (see Appendix A). This protocol refrains unwanted complex behavior leading to higher dissipation [31] while profiting from ferroelectric self-charging characteristics.

## 4. Conclusions

An innovative all-solid-state coaxial structural battery concept for beam-like applications was proposed. This alternative design of structural batteries was based on the use of a safe and non-flammable ferroelectric Na^+^-electrolyte blended with a polyvinyl acetate glue. The compound maintained all the electrochemical properties observed in the pure electrolyte double-layer pouch cells with asymmetric electrodes, such as the long-lasting discharges and cycling, enhanced thermal properties, self-charging, and self-oscillation. 

The structural beam concept was achieved using a CFRP outer shell for hosting the cathode/collector; the inner volume was filled instead with the structural electrolyte and a coaxial anode/collector rod. The output voltage of the cell is based on the chemical potential difference of the copper and the aluminum (alumina surface layers), which are discharged as cathode and anode, respectively. No other chemical reactions were utilized to extend the output voltage of the cell (e.g., using MnO_2_ deposited in the cathode) which may configure an additional option. The beam-like power composite reached an energy density of 989 Wh kg^−1^ for the positive electrode only and an energy density of 78.1 Wh kg^−1^ and specific energy of 86.0 Wh L^−1^ for the active electrolyte (Na_2.99_Ba_0.005_OCl), that drops to 38.0 Wh kg^−1^ and 56.2 Wh L^−1^ for the whole cell including the CFRP structural element.

Greater thermal properties showed how this cell could be used for harvesting thermal energy from the residual heat or higher temperatures built out of industrial, households, engines, and databanks processes. The latter capacity does not lean on a temperature gradient as a thermoelectric or on a temperature rate as a pyroelectric material.

This work aims to introduce an alternative all-solid-state coaxial structural battery concept with improved capacitance and to show its implementation for beam-like structures. Its coaxial shape makes this configuration extremely appealing for any beam-like based structure or vehicles where the volume constraints are challenging. The cells can be associated in series and show no dimensional constraints. 

Mechanical tests demonstrated the possibility to improve the mechanical bending load of a hollow carbon fiber reinforced plastic (CFRP) beam while working like a battery. The results highlight not only an increase in the peak load but also a pseudo-ductile bending failure. Simultaneously, the battery demonstrated its robustness in bearing severe mechanical deformation without compromising its functionalities and safety.

A cycling protocol for practical purposes allows for a 3 min charge to 24 h discharge at a constant voltage.

It is envisioned that coaxial cells can be further engineered to increase the specific energy by making the aluminum rod hollowed and by reducing the amount of electrolyte by tailoring the relative radius of the electrodes/current collectors. Currently, this novel coaxial cell possesses the highest specific energy among the structural safe cells, and it is easier to fabricate, cheaper, Na^+^-based and all-solid.

## Figures and Tables

**Figure 1 molecules-26-05226-f001:**
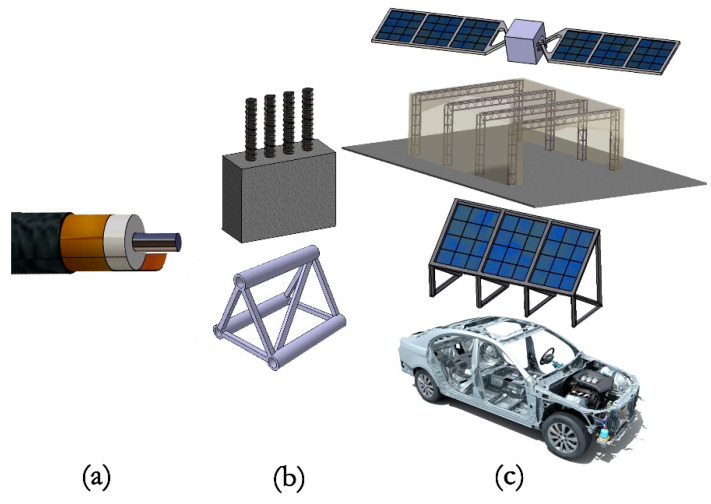
The coaxial all-solid-state battery and possible applications; (**a**) coaxial structural battery; (**b**) structural components: rebars, truss sets; (**c**) structures: vehicles, industrial equipment, solar panels, satellite arms, and antennas.

**Figure 2 molecules-26-05226-f002:**
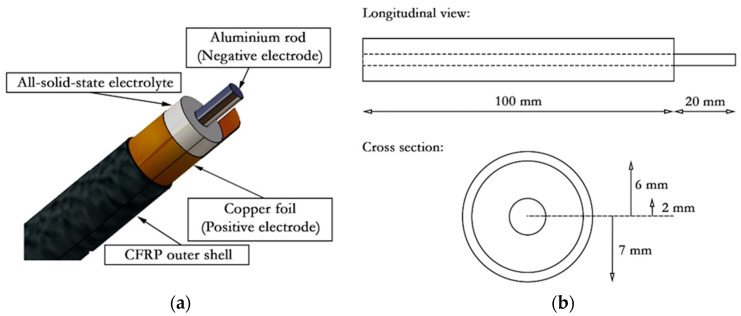
Coaxial structural battery design; (**a**) main components, (**b**) views with dimensions. The coaxial structural battery is composed of an external carbon fiber reinforced plastic (CFRP) shell co-cured with a thin copper film that works as the positive electrode, a Na^+^-based all-solid-state ferroelectric electrolyte, and a concentric aluminum rod as the negative electrode.

**Figure 3 molecules-26-05226-f003:**
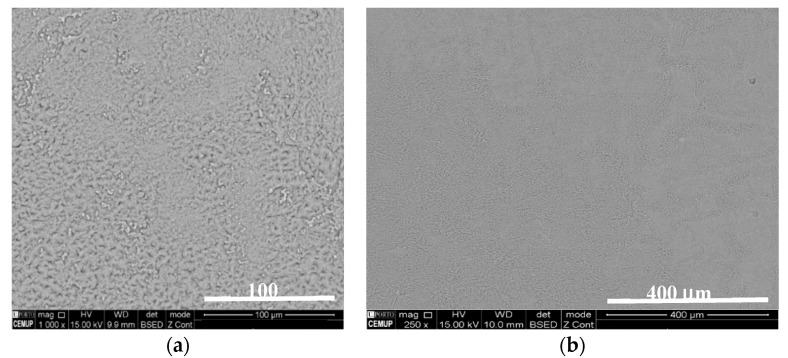
Ferroelectric composite (80% Na-glass electrolyte + 20% polyvinyl acetate PVAc-C_4_H_6_O_2_) SEM microphotographs (backscattered radiation). Magnification: (**a**) 250×; (**b**) 1000×.

**Figure 4 molecules-26-05226-f004:**
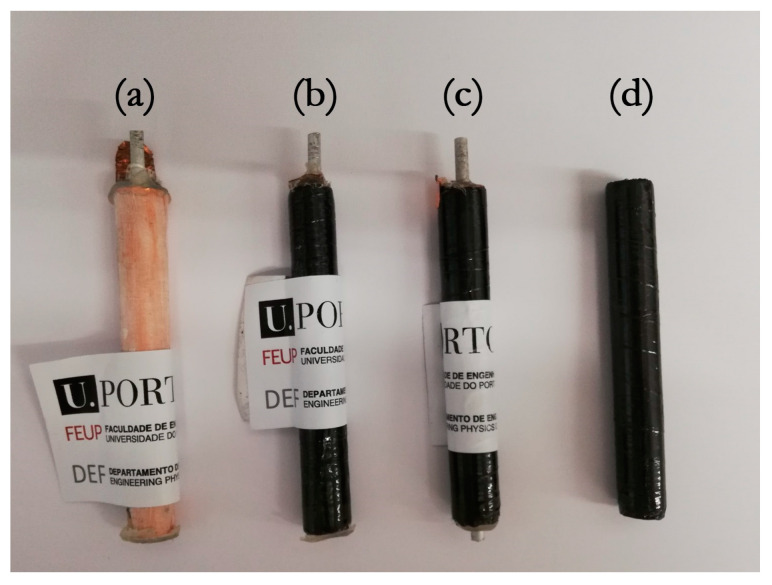
Showcase of the type of coaxial cells and parts investigated; (**a**) structural battery with thin glass fiber shell such as schematized in Figure 2a; (**b**) and (**c**) two coaxial cells that were analised in this study; (**d**) CFRP shell.

**Figure 5 molecules-26-05226-f005:**
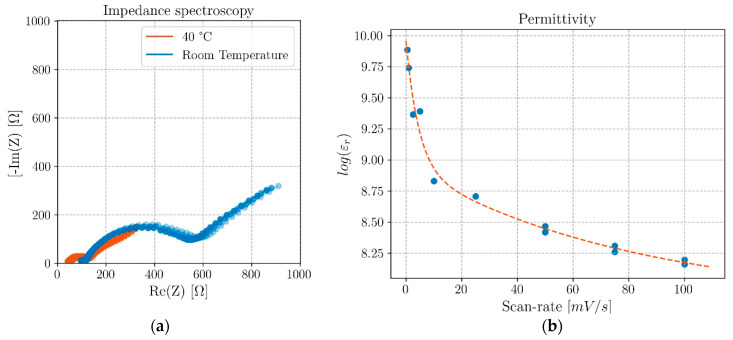
Electrochemical cell analyses; (**a**) Impedance spectroscopy at different temperatures obtained by EIS; (**b**) Permittivity at 40 °C obtained by cyclic voltammetry, CV cycles, for differ-ent scanning rates and fitted by a two-phase exponential decay curve.

**Figure 6 molecules-26-05226-f006:**
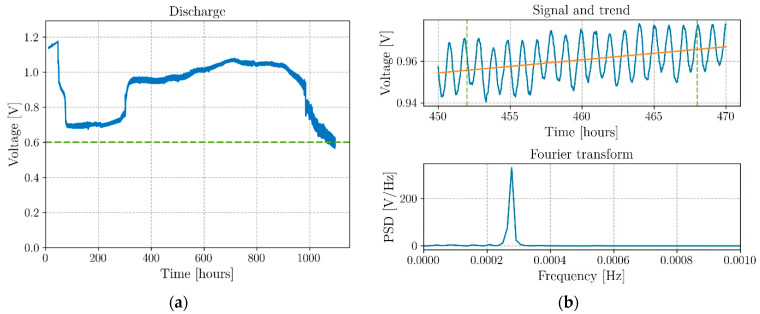
Typical discharge of an all-solid-state coaxial structural battery; (**a**) full discharge with 0.98 kΩ external resistor, corresponding to ~1 mA, highlighting self-charge. The resistor was connected after 47 h; the cell started to self-charge after 84 h at 0.70 V; a step self-charge from 0.75 to 0.96 V occurred after 300 h, giving rise to a smooth self-charge of 27 days (648 h) at <V> = 1.00 V, for a maximum of 1.07 V at day 29 (715 h); (**b**) self-oscillation detail and trend line extracted between 450 and 470 h during the test with fast Fourier transform analysis detecting a characteristic period of 1 hour.

**Figure 7 molecules-26-05226-f007:**
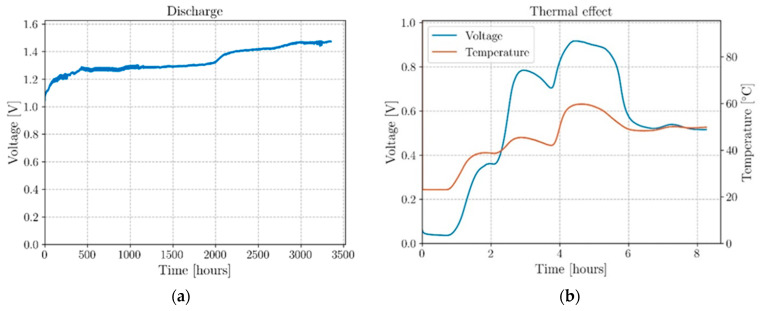
Relevant discharge curves. (**a**) Discharge curve with a 26.6 kΩ material resistor with voltage increase from 1.03 to above 1.47 V in >140 days; (**b**) Voltage increase while discharging with a 1.8 kΩ material resistor is induced by an imposed rising temperature. A temperature increase reflects a sharp increase in the permittivity and ionic conductivity of the electrolyte.

**Figure 8 molecules-26-05226-f008:**
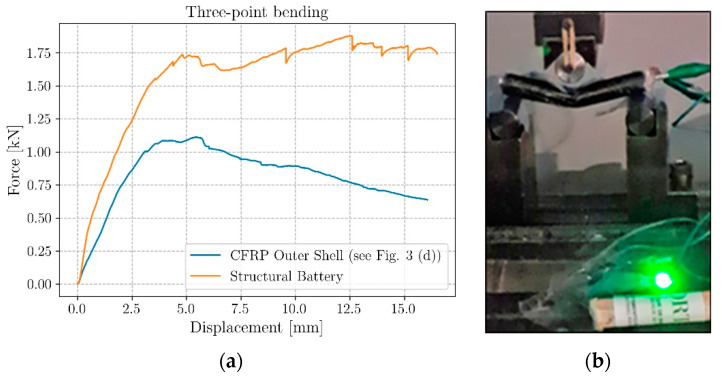
Three-point bending test; (**a**) Force vs. displacement curves for a structural battery and the CFRP outer shell only, (**b**) structural battery lighting an LED while the mechanical test is performed.

**Figure 9 molecules-26-05226-f009:**
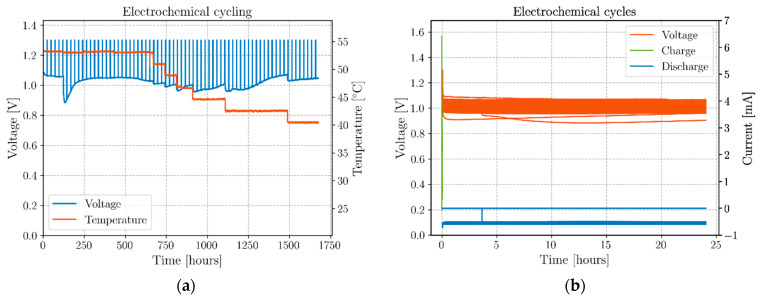
Post-mortem electrochemical cycling (70 cycles). Three-minute charge at a constant voltage of 1.3 V followed by 24 h discharges with a 1.8 kΩ material resistor (~0.6 mA) not exhibiting self-cycling. (**a**) Full charge/discharge sequence; (**b**) single cycles’ details for 70 cycles (Voltage and Charge and Discharge currents).

**Table 1 molecules-26-05226-t001:** All-solid-state structural battery performance.

Authors		Electrochemical Cell Components	Manufacturing	Specific Energy(Wh/kg)	Discharge Voltage (V)
Ladpli et al.	[16]	NMC on Al foil (60/10 μm) Graphite on Cu foil (96/10 μm) LiPF_6_ in EC/DMC/DEC on trilayer polyolefin separator (PP/PE) (25 μm) Liquid electrolyte Not safe	CFRP shell with interlocking rivets with embedded customized cell	123–139 (No rivets)	3.7
Asp et al.	[23]	Carbon fibers (65 μm) LiFePO_4_ on Al foil (50/30 μm) 0.4 M LiBoB/0.6 M LiTf on Whatman GF/A separator (185 μm) ionic liquid (Safe, low ionic conductivity)	Cell components stacking in a pouch bag Structural electrolyte addition Sealing and curing	**11.6**	2.8
Asp et al.	[23]	Carbon fibers (125 μm) LiFePO_4_ on Al foil (50/30 μm) 0.4 M LiBoB/0.6 M LiTf on glass fiber separator (70 μm) ionic liquid Safe, low ionic conductivity	Cell components stacking in a pouch bag Structural electrolyte addition Sealing and curing	**23.6**	2.8
Moyer et al.	[24]	LiFePO_4_ on carbon fiber Graphite on carbon fiber LiTFSI in EMIMBF_4_ on Whatman glass fiber separator ionic liquid Safe, low ionic conductivity	Battery layup in carbon fiber shells Epoxy resin impregnation and curing	**35**	N.A.
Moyer et al.	[25]	LiFePO_4_ on carbon fiber Graphite on carbon fiber LiPF_6_ in EC/DEC Celgard 2525 separator Liquid electrolyte Not safe	Battery layup in carbon fiber shells with PAN-coated interfaces Epoxy resin impregnation and curing	52 (max. 58)	2.5
Pyo et al.	[26]	LiFePO_4_ on Al foil, Li_4_Ti_5_O_12_ on Al foil, LiPF_6_ in EC/DMC on Double-layer glass fabric 2116 separator Liquid electrolyte with LTO anode but still not fully safe	Tubular structure and battery components stacking and curing PDMS application and curing Electrolyte filling Final sealing	**3** (One tubular laminated composite battery)	1.8

**Table 2 molecules-26-05226-t002:** All-solid-state coaxial structural battery performance.

		Full Cell	Full Cell No CFRP	CFRP Shell	Positive Electrode (Cu)	CFRP + Cu	Na+-Electrolyte + PVac (80%/20%)
Mass	(g)	21.6	15.2	6.4	0.83	7.22	10.5
Specific capacity	(mAh/g)	41.0	58.1	-	1067	123	82.2
Energy density	(Wh/kg)	**38.0**	75.8	-	989	114	78.1
Specific energy	(Wh/L)	56.2	76.3	-	-	-	83.9
Cost	(€/kWh)	<1200 ^a^	<10	-	-	-	-

^a^ The CFRP used in these structural batteries is suitable for aerospace and other high-performance structures. An inexpensive outer shell may be used in industrial or household applications.

## Data Availability

The data is available upon request.

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
