# Peer review of "An All-Solid-State Coaxial Structural Battery Using Sodium-Based Electrolyte"

_molecules, 2021, doi:10.3390/molecules26175226_

Round 1

Reviewer 1 Report

From my point of view, the publication of the ‘’article (An all-solid-state coaxial structural battery using the sodium-based electrolyte.) can only be considered after major revisions.

In the introduction, the part should be improved and the author would be discussed more the coaxial structural batteries. Re-write the last two paragraphs to clear the concept of the article to the reader.

The materials and method part is very organized and well written.

The result and discussion part need to improve especially just not put experimental value but also explain the result.

Author Response

The authors thank reviewer #1 for her/his valuable comments and work.

Reviewer #1

  1. In the introduction, the part should be improved and the author would be discussed more the coaxial structural batteries. Re-write the last two paragraphs to clear the concept of the article to the reader.

R1. The present coaxial all-solid-state batteries configure a novel concept.

The introduction was reviewed and Table 1 with the comparison between the different features among the latest structural batteries, was added. Additionally, we have highlighted all the features that make the present technology much more appealing and pointed out its structure and properties.

  1. The materials and method part is very organized and well written.

R2. Thank you.

  1. The result and discussion part need to improve especially just not put experimental value but also explain the result.

R3. Additional discussion was introduced to clarify and point to the very interesting features of the theoretical aspects as well as applications.

Reviewer 2 Report

In this work,the author design novel coaxial all-solid-state battery, and test cell  electrochemical,I wonder how does this all solid state battery compare with other designs?Please add the performance comparison. Furthermore, It is better to add some characterization analysis of microstructure,  the change of microstructure has an important impact on electrochemical and mechanical properties.

Author Response

The authors thank reviewer #2 for her/his valuable comments and work.

Reviewer #2

  1. In this work,the author design novel coaxial all-solid-state battery, and test cell electrochemical,

I wonder how does this all-solid-state battery compare with other designs? Please add the performance comparison. Furthermore, It is better to add some characterization analysis of microstructure,  the change of microstructure has an important impact on electrochemical and mechanical properties

R1. The present coaxial all-solid-state batteries configure a novel concept.

The introduction was reviewed and Table 1 with the comparison between the different features among the latest structural batteries, was added.

Additionally, we have highlighted all the features that make the present technology much more appealing and pointed out its structure and properties. We have made a comparison of the performance among structural batteries, cylindrical batteries, and photovoltaic cells (in regards to harvesting).

We have added SEM analyses showing the microstructure of the composite and XPS of the PVAc glue.

Reviewer 3 Report

The manuscript presents a coaxial structural battery with sodium-based all-solid-state ferroelectric electrolyte, and their charge/discharge properties. Overall, the article’s topic is interesting, although, in its current form, it appears too preliminary for publication. The reviewer suggests that this manuscript could be resubmitted to Molecules after describing the principles and processes of electrochemical testing clearly. Meanwhile, clearing the intrinsic reasons of the experimental results, rather than an experimental report. What’s more, eliminating some low-level errors (eg. …Figure S3 and if Figure 6a respectively. From the analysis of the results is it possible…).

Author Response

The authors thank reviewer #3 for her/his valuable comments and work.

Reviewer #3

  1. The manuscript presents a coaxial structural battery with sodium-based all-solid-state ferroelectric electrolyte and their charge/discharge properties. Overall, the article’s topic is interesting, although, in its current form, it appears too preliminary for publication. The reviewer suggests that this manuscript could be resubmitted to Molecules after describing the principles and processes of electrochemical testing clearly.

R1. We have tried to clarify the principles and electrochemical processes occurring with this complex system based on a ferroelectric electrolyte and cells of thereof demonstrating competing phenomena not always comparable to a traditional battery.

Additional data was added to the manuscript and testing elements were described in more detail. We have added SEM analyses showing the microstructure of the composite and XPS of the PVAc glue.

  1. Meanwhile, clearing the intrinsic reasons of the experimental results, rather than an experimental report. What’s more, eliminating some low-level errors (eg. …Figure S3 and if Figure 6a respectively. From the analysis of the results is it possible…).

R2. We have clarified the phenomena, improved the manuscript, and corrected the typos and errors.

Round 2

Reviewer 1 Report

The authors have addressed all of my concerns and the manuscript has been significantly improved to warrant publication.

Reviewer 2 Report

it can be accepted